# Coated Hematite Nanoparticles Alleviate Iron Deficiency in Cucumber in Acidic Nutrient Solution and as Foliar Spray

**DOI:** 10.3390/plants12173104

**Published:** 2023-08-29

**Authors:** Amarjeet Singh, Fruzsina Pankaczi, Deepali Rana, Zoltán May, Gyula Tolnai, Ferenc Fodor

**Affiliations:** 1Department of Plant Physiology and Molecular Plant Biology, ELTE Eötvös Loránd University, Pázmány Péter Lane 1/c, 1117 Budapest, Hungary; amarpc@student.elte.hu (A.S.); pankafru@gmail.com (F.P.); deepalirana944@gmail.com (D.R.); 2Doctoral School of Biological Sciences, ELTE Eötvös Loránd University, Pázmány Péter Lane 1/c, 1117 Budapest, Hungary; 3Doctoral School of Environmental Sciences, ELTE Eötvös Loránd University, Pázmány Péter Lane 1/a, 1117 Budapest, Hungary; 4Institute of Materials and Environmental Chemistry, Research Centre for Natural Sciences, Eötvös Loránd Research Network, Magyar Tudósok Blvd. 2, 1117 Budapest, Hungary; may.zoltan@ttk.mta.hu; 58/A Kondorosi Street, H-1116 Budapest, Hungary; gyula.tolnai@yahoo.com

**Keywords:** iron, micronutrient, nanofertilizers, nanoferrihydrite, nanohematite, XRF mapping

## Abstract

Micronutrient iron (Fe) deficiency poses a widespread agricultural challenge with global implications. Fe deficiency affects plant growth and immune function, leading to reduced yields and contributing to the global “hidden hunger.” While conventional Fe-based fertilizers are available, their efficacy is limited under certain conditions. Most recently, nanofertilizers have been shown as promising alternatives to conventional fertilizers. In this study, three nanohematite/nanoferrihydrite preparations (NHs) with different coatings were applied through the roots and shoots to Fe-deficient cucumber plants. To enhance Fe mobilization to leaves during foliar treatment, the plants were pre-treated with various acids (citric acid, ascorbic acid, and glycine) at a concentration of 0.5 mM. Multiple physiological parameters were examined, revealing that both root and foliar treatments resulted in improved chlorophyll content, biomass, photosynthetic parameters, and reduced ferric chelate reductase activity. The plants also significantly accumulated Fe in their developing leaves and its distribution after NHs treatment, detected by X-ray fluorescence mapping, implied long-distance mobilization in their veins. These findings suggest that the applied NHs effectively mitigated Fe deficiency in cucumber plants through both modes of application, highlighting their potential as nanofertilizers on a larger scale.

## 1. Introduction

Iron (Fe) is an essential micronutrient for plants, playing a vital role in various metabolic processes, such as photosynthesis, respiration, and chlorophyll biosynthesis. It acts as an electron donor or acceptor in enzymes in the form of Fe–sulfur clusters, heme, and free Fe ion [1]. As a constituent of the photosynthetic electron transport chain, Fe plays a crucial role in plant growth and development [2].

Despite being the fourth most abundant element in the Earth’s crust, plants continue to suffer from Fe deficiency [3]. Fe deficiency is a common problem in agriculture, particularly in alkaline or calcareous soils where higher pH levels prevent the uptake of Fe [4]. Fe deficiency manifests in plants as chlorosis or yellowing of the leaves, stunted growth, and reduced crop yields. The symptoms are more pronounced in young leaves and can result in poor root development, reduced yield, or even complete crop failure [5]. Fe deficiency can also reduce the production of proteins involved in the photosynthetic apparatus, decreasing photosynthetic efficiency. This includes alterations in the thylakoid membrane structures, photosynthetic electron transport chain, and reduced formation of iron–sulfur complexes [6]. Chloroplast biogenesis and differentiation are also negatively affected by Fe deficiency [7].

Plants have evolved two main strategies to acquire Fe from the soil. Dicots and non-graminaceous monocots use reduction-based strategy I. In this strategy, Fe is reduced from the ferric (Fe^3+^) to ferrous (Fe^2+^) form by a membrane-bound ferric reductase oxidase (FRO). The reduced Fe is then transported into the root epidermal cells via another membrane-localized iron-regulated transporter (IRT1). Phenolics, particularly coumarins and other secondary metabolites such as flavins, are also secreted by the roots and can directly reduce Fe^3+^ and chelate both Fe (III) and Fe (II), thus improving Fe mobilization and reduction [8]. Strategy II is based on chelation of Fe and is used by graminaceous plants. It is dependent upon the release of mugineic acid family phytosiderophores (MAs) in the rhizosphere that chelate ferric iron. In rice, transporter of mugineic acids (TOM1) is involved in the secretion of deoxy mugineic acid (DMA) from the root to the rhizosphere in Fe deficiency [9]. The chelated Fe-phytosiderophore complex is then transported into the root via yellow stripe (YS) and yellow stripe-like (YSL) transporters [10,11].

The traditional methods for alleviating Fe deficiency in agriculture include soil amendment with Fe-rich fertilizers, e.g., Fe sulfate, Fe chelate, or Fe oxide. In addition, use of Fe-rich organic matter, such as compost or green manure, is also considered. However, these have certain limitations. For example, Fe-rich fertilizers are usually ineffective in soils with high pH levels, as Fe is less soluble in alkaline conditions [12]. Moreover, even though they are cost-effective, higher amounts need to be applied in soil, which can prove to be toxic to plants and cause leaf damage or stunted growth. Fe fertilization with these may provide a temporary boost to Fe levels in plants, but the effects are not long-lasting and repeated applications may be necessary [13,14]. In addition, poor distribution of these fertilizers may result from leaching or fixation in soil, thus decreasing their effectiveness. On the other hand, chelated fertilizers, such as Fe-EDDHA, are effective even in alkaline conditions, but they are uneconomic and their application is limited to cash crops [15].

Due to the limitations of conventional fertilizers, there is increasing interest in exploring more environmentally friendly solutions, such as manufactured nanomaterials. Nanomaterials have unique properties, such as high surface area and reactivity, making them ideal for plant nutrient supplementation. The use of nanoscale Fe particles can provide a more efficient and sustainable solution for Fe supplementation, as they can be more easily absorbed by plants, reducing the amount of total fertilizer required [16]. Additionally, the use of nanoscale Fe particles can help to alleviate the issue of soil alkalinity, as they can be designed to release Fe at a specific pH level, improving the availability of Fe for plants [17]. In agriculture, Fe-based nanomaterials have been studied for their potential to alleviate Fe deficiency in crops. Recent studies have demonstrated that the application of nanofertilizers, both through the root system and as foliar treatment, can dramatically enhance plant growth and yield [18,19,20].

Fe exists as four main different forms in soils: (1) as Fe^2+^ released from primary minerals such as silicates, borates, and sulfates; (2) as ferric oxyhydroxides after oxidation and precipitation under aerobic conditions within the pH range of 5 to 8 (also known as pedogenic Fe); (3) as soluble and exchangeable Fe; and (4) as short-range ordered crystalline minerals, including ferrihydrite and schwertmannite, which are bound to organic matter [21]. Fe oxides that are most abundant in soil with low solubility include geothite (α-FeOOH) and hematite (α-Fe_2_O_3_) [16,22].

Fe oxides are known to have high affinity for water, with the ability to absorb up to one mole of excess H_2_O per mole of Fe oxide. The affinity of these oxides for water is further increased by their small particle sizes, particularly at the nanoscale. In fact, nanoparticles of hematite can form naturally in soil through nucleation when groundwater becomes saturated with clusters of the same mineral phase. These nanoparticles are an important source of bioavailable Fe, which is essential for plants and microorganisms to thrive, grow, and diversify [23,24,25].

In previous work, we compared various aspects of nanoferrihydrite and nanohematite suspensions, including their utilization in providing Fe to plants [26]. It was found that nanohematites performed better than nanoferrihydrites, and surfactants such as polyethylene glycol polymers may increase the efficiency of nanohematites. In the present investigation, nanocolloid suspensions of hematites/ferrihydrites with different surfactant coatings were assessed for their effectiveness as supplements in nutrient solutions and as foliar sprays to address Fe deficiency in plants. By using these nanohematite/nanoferrihydrite (NH) preparations, this study sought to identify the most effective application method to enhance plant growth and alleviate Fe deficiency symptoms in cucumber model plants.

## 2. Results

### 2.1. Greening and Chlorophyll Concentration

In this study, three NH colloid suspensions differing mainly in the surfactant (PEG-1500, NH-S1; Emulsion 104D, NH-S2; and SOLUTOL HS 15, NH-S3) used for stabilization were used as amendments to nutrient solutions or as foliar fertilizers. For further characterization of the NHs, please see Section 4.2.

The roots of 2-week-old Fe-deficient (dFe) cucumber plants were supplied with a nominal concentration of 20 µM Fe in NH-S1-3 at two different pH values (unbuffered acidic pH 6.0 and alkaline pH buffered at 8.5). Plants at acidic pH showed a remarkable increase in chlorophyll (Chl) a+b content (Table 1). A significant 20–21-fold increase was observed for all three NHs in comparison with dFe plants. The Chl a/b ratio decreased when plants were supplied with NHs compared to dFe plants. Greening was also assessed using a SPAD instrument. Within 24 h of NH supplementation, plants showed an increase in the SPAD index. All three NH-treated plants exhibited a 13–14-fold increase in the SPAD values of the 1st leaf after 6 days of treatment compared with the corresponding dFe plants. For the 2nd leaves as well, in comparison to dFe plants, the SPAD values showed an 11-fold increase. In general, the relative SPAD values for dFe leaves continued to decline with time (Figure 1a,b). At pH 8.5, there was no greening at all and the Chl a/b ratio did not change significantly either. (SPAD values are not shown).

In a preliminary experiment, it was found that NHs applied alone as foliar treatment did not induce Chl synthesis. For this reason, we applied citric acid (Cit), ascorbic acid (Asa), and glycine (Gly) as pre-treatments before the application of NHs. The leaves of plants with foliar application of NH-S1-3 demonstrated remarkable greening regardless of the pre-treatment received. The SPAD values of both the 1st and 2nd leaves increased significantly, ranging from 1.3- to 3-fold compared to their original values (Figure 1c,d) for all NHs. However, the greening of leaves was not uniform across the blade, with small, dark green spots corresponding to NH droplets clearly visible on the lamina (Appendix A). Nevertheless, with time, the young 2nd leaves exhibited subsequent growth in surface area and increased overall greening.

### 2.2. Biomass and pH

Plants that were supplied with NHs in nutrient solutions at acidic pH showed an increase in biomass after 6 days of treatment (Figure 2). NH-S2-treated plants showed maximum gain in biomass, with a significant 2-fold increase in dry weight (DW) upon comparison with dFe plants. Importantly, the biomass recorded for this treatment was higher than that of Fe-citrate-treated control plants. The other two NHs also resulted in enhanced biomass compared to dFe plants, in the order NH-S2 > NH-S3 = NH-S1, but the change was not significant. For plants grown at alkaline pH, no significant change in biomass was recorded when compared with dFe plants (data not shown).

Regarding the pH of the unbuffered nutrient solution, there was smaller decrease in the pH values of plants treated with NH-S1-3 at the end of experiment than that of dFe plants (Table 2). As anticipated, the pH of the dFe plants decreased by approximately one pH unit. The final pH difference between nutrient solutions of the NH-treated and untreated dFe plants was significant whereas the 3 NHs were not different from each other. For plants at alkaline pH, the final pH of the nutrient solution did not deviate significantly from the original pH 8.5 as it was maintained by buffer (data not shown).

The pH of the nutrient solution of plants with foliar treatment, as an indicator of the downregulation of Fe-efficiency reactions, was slightly influenced by the type of pre-treatment they received. For all NHs, pre-treatments did not result in highly remarkable changes in the pH of the nutrient solution. Pre-treatment with Cit and in case of NH-S3 with Asa seemed to result in higher pH values than the other treatments. Surprisingly, the pH of plants pre-treated with Gly was remarkably low and for NH-S2, it was even lower than that of the dFe plants.

### 2.3. Ferric Chelate Reductase Activity

As the plants grown at alkaline pH did not show promising greening, further measurements were performed only for plants treated with nutrient solutions at unbuffered pH and with foliar sprays. Three days after treatment, the root ferric chelate reductase (FCR) activity was measured to evaluate the regeneration from Fe deficiency. The results shown in Figure 3a clearly demonstrate that the FCR activity of all NH-treated roots was significantly lower than that of dFe roots. The FCR activity of the roots of treated plants was in the range of that of sFe plants grown with Fe-citrate.

The root FCR activity of the plants supplied with NHs as foliar spray after 3 days of treatment is shown in Figure 3b. All NH treatments resulted in significant suppression of FCR activity (except for NH-S1/Gly), suggesting that the plants could utilize Fe from the supplied NHs. Interestingly, the pre-treatment spraying solution had a significant impact modulating the effect of the NHs. The lowest FCR activity for all NHs was recorded in plants pre-treated with Cit, with significant 4.62-, 3.31-, and 3.02-fold decreases after NH-S3, NH-S2, and NH-S1 treatments, respectively, compared to the FCR activity of dFe roots. Similarly, Asa pre-treatment caused a significant suppression of root FCR activity for all three NHs, with 1.65-, 3.21-, and 1.88- fold decreases measured for NH-S3, NH-S2, and NH-S1, respectively. In contrast, Gly pre-treatment had a relatively lesser effect on FCR activity, with a 2-fold decrease only for NH-S3 and 1.7-fold decrease for NH-S2. For NH-S1, Gly pre-treated plants had comparable FCR activity to that of dFe plants. Therefore, these findings implied that pre-treatment of plants with Cit, Asa, and Gly had a significant repressive effect on root FCR activity for all three NHs, except for Gly pre-treatment in NH-S1-treated plants.

### 2.4. Photosynthetic Efficiency

Higher values of CO_2_ assimilation rate (A_net_) observed in Fe-citrate control and NH-treated plants indicated their greater ability to photosynthesize and convert CO_2_ into organic carbon compounds (Table 3). However, a negative A_net_ value was recorded for dFe plants, indicating release instead of fixation of CO_2_.

The efficiency of the photosystem in fixing CO_2_ was assessed by measuring ΦCO_2_, which represents the quantum yield of CO_2_ fixation during photosynthesis. The dFe plants exhibited significantly lower ΦCO_2_ values compared to Fe-citrate control and NH-treated plants, indicating poor performance in fixing the incoming CO_2_ (Table 3). This reduced CO_2_ fixation was attributed to the observed lower biomass as well as the reduced plant growth and productivity in dFe plants. Foliar spray-treated plants also showed a significant increase in ΦCO_2_ (0.0262 ± 0.0070 µmol µmol^−1^ photons) compared to dFe plants (0.00499 ± 0.00214 µmol µmol^−1^ photons).

The maximum quantum efficiency of photosystem II (Fv/Fm) of plants supplied with NHs in nutrient solution was in the healthy range compared to sFe plants (Table 3). Similarly, NH-S1 applied via the foliar method, regardless of pre-treatment, as well as NH-S2 with Cit pre-treatment caused full recovery. NH-S2 with Asa and Gly and NH-S3 with all pre-treatments also caused a significant increase in the Fv/Fm values of leaves compared to dFe plants; however, they did not reach the optimal range.

Similarly, the actual quantum efficiency of photosystem II (PSII) in a light-adapted state (Fv’/Fm’) showed recovery for all NH-treated dFe cucumbers, as the values reached the same levels as Fe-citrate control plants, with a 68% recorded increase compared to dFe plants (Table 3). Additionally, for all foliar treatments, a significant average increase of 88% in Fv’/Fm’ (0.582 ± 0.044) was observed compared to dFe plants, which approached that of Fe-citrate control plants.

The actual quantum efficiency of photosystem II (ΦPSII) for all plants (root-applied NHs) was in the same range as that of Fe-citrate control plants (0.47 ± 0.028) (Table 3). The ΦPSII value was recorded to be much lower for dFe plants, which was 80% less than that of Fe-citrate control plants. Additionally, for foliar spray-treated plants, the ΦPSII values increased significantly, but slightly lower values were recorded for NH-S3 than for the other two NHs.

In the presence of Fe deficiency stress, the non-photochemical quenching (NPQ) values demonstrated a marked reduction, indicating an impairment of both electron transport and proton pumping, which are critical for photosynthesis (Table 3). Fe-deficient plants, therefore, face a challenge in curbing the excess energy that may lead to photodamage and reduced photosynthetic efficiency. However, plants treated with NHs, both via foliar and root application, as well as Fe citrate control plants, exhibited NPQ values that were comparable in range. This suggested that they effectively dissipated excess energy and potentially experienced photoprotection under conditions of stress.

### 2.5. Element Analysis

The 2nd leaves of plants supplied with NHs in nutrient solution were analyzed for Fe, Mn, and Zn concentrations. The NH-treated plants successfully internalized the Fe originating from NHs, as shown in Figure 4. The Fe concentration doubled in NH-treated plants, but it did not reach that of sFe plants. This was accompanied by a visible but yet non-significant reduction in the uptake of Mn and Zn. The levels of Mn and Zn rose more than 2.3- and 2.6-fold in dFe plants in the absence of Fe in comparison with Fe-citrate control plants.

### 2.6. X-ray Fluorescence Mapping of the Leaves

X-ray fluorescence mapping was performed on arbitrarily chosen sections of the 2nd leaves of plants to observe the distribution of Fe in comparison with well-detectable macroelements such as K and Ca (Figure 5). The macroelements showed well-characterized distributions in the leaves, with high intensity. In sFe plants, Ca showed accumulation especially in small-concentrated patches that were evenly distributed and in the veins. In dFe plants, the distribution was similar but the intensity was much lower, indicating lower concentrations. Potassium showed more uniform distribution in the interveinal sections, but its accumulation was confined to the major and minor veins in both sFe and dFe plants. This distribution was primarily observed in plants after application of foliar treatment. Iron accumulation was seen in sFe plants but not in untreated dFe plants. Patches of Fe were also seen on NH-S3/Cit and NH-S3/Gly leaves, obviously due to treatment solution dried on the surface, but not in other cases. A well-detectable Fe intensity was seen in NH-S1/Cit, NH-S1/Gly, NH-S2/Cit, and NH-S2/Asa, whereas in other treatments, the Fe intensity was much lower although the amount of Fe applied in the NH treatments was the same.

## 3. Discussion

In this investigation, three NH colloid suspensions with different coatings, when supplemented in unbuffered nutrient solution at a nominal concentration of 20 µM, were found to ameliorate Fe deficiency symptoms in cucumber plants. The treated dFe plants showed enhancements in biomass, chlorophyll content, and improved photosynthetic efficiency. All of these indicate a metabolic shift towards Chl synthesis and assimilation of the photosynthetic apparatus. Furthermore, the acidification of the nutrient solution and FCR activity of the root tips decreased, which showed downregulation of the high-affinity Fe uptake system [27]. Element analysis of the 2nd leaves confirmed that the plants were able to efficiently take up and utilize Fe from all three NHs at unbuffered pH. On the contrary, when the NHs were supplied in nutrient solutions buffered to pH 8.5, no improvements were found in the measured parameters and the plants remained Fe deficient. Various studies have been carried out to understand the effect of hematite particles on plants. Boutchuen et al. [18] applied hematite NPs of ∼16 nm at low (0.022 g·L^−1^ Fe) and high (1.1 g·L^−1^ Fe) concentrations to seeds of four different commercial crops, resulting in 230–280% increases in plant growth. The treatment also increased the survival span of plants, doubled fruit production per plant, accelerated fruit production by nearly two times, and produced healthy second-generation plants with slight species-specific variations. In a recent study by Ndou et al. [28], green-synthesized hematite NPs (average NP diameter of 18 nm) positively impacted sorghum bicolor growth and inhibited the oxidative damage of proteins, DNA, and lipids by enhancing nutrient uptake and osmoregulation under drought conditions. Rath et al. [29] also demonstrated enhanced seed germination for *Triticum aestivum*, fenugreek (*Trigonella foenum-graecum*), Bengal gram (*Cicer arietinum*), and broad bean (*Vicia faba*) in the presence of a low amount of hematite NPs (20 mg·L^−1^). Similarly, Pariona et al. [20] studied the effect of hematite NPs (100 nm, ovoid and rounded shape) on *Zea mays* and reported enhanced growth and chlorophyll content of maize seedlings. All of these results are in accordance with the positive effects of NHs found in our study.

We also studied the effectiveness of the three NHs as foliar fertilizers applied on dFe model plants. However, in a preliminary experiment, it was found that NHs applied on the leaves alone was not effective at all. Therefore, we examined the ability of three different pre-treatments (Cit, Asa, and Gly) to facilitate Fe uptake from the supplied NHs.

Various studies have explored the biostimulatory effects of organic and amino acids on plant growth and nutrient uptake enhancement under various biotic and abiotic stress conditions [30,31,32]. Rasp [33] found that foliar spraying of amino acids and FeSO_4_ was more effective in promoting the absorption, translocation, and utilization of Fe in grapevine plants compared to FeSO_4_ application alone. Additionally, studies by Tejada and Gonzalez [34] with asparagus (*Asparagus officinalis* L.) plants demonstrated the beneficial effects of amino acids, including increased root and shoot growth and chlorophyll content in leaves. Sánchez-Sánchez et al. [35,36] also observed that soil application of amino acids mixed with Fe-EDDHA improved Fe uptake and several fruit quality parameters in citrus and tomato plants. Most recently, Rajaie and Tavakoly [37] found that citric acid and sulfuric acid enhanced the leaf chlorophyll content of orange trees, indicating their role in the remobilization of already existing Fe in leaves. The improvement of plant Fe nutrition could be attributed to the ability of amino acids to act as natural chelators and efficient carriers of Fe into plants, as well as their effects on metabolism and plant physiology, such as the stimulation of H^+^-ATPase and FCR activity and increased cell membrane permeability, as suggested by Sánchez-Sánchez et al. [35,36].

When we applied Cit, Asa, or Gly as a pre-treatment, it was found that Cit was the most effective regardless of the NH used. Plants treated with Cit had the lowest FCR activity and higher Fe accumulation compared to those treated with Gly or Asa. It is well established that Cit, succinic acid, and malic acid are involved in Fe metabolism [37]. Cit, specifically, serves as an intermediate in the Krebs cycle and forms a crucial Fe (III)–citrate complex that is highly mobile and facilitates the transport of Fe between various plant compartments over long distances [38,39]. The applied Cit may mobilize and complex ferric Fe from the applied NHs. Thus, Cit proved to be an effective pre-treatment solution for the NH foliar fertilizers, enhancing Fe utilization by the leaves and leading to the downregulation of root FCR activity in our model plants.

Another compound we applied as a pre-treatment, Asa, is a well-known acid for its ability to effectively scavenge oxidative stress and reduce the production of free radicals caused by abiotic stress factors [40]. Applying Asa to the leaves of plants can stimulate the plant’s natural production of Asa, helping to mitigate stressful conditions [41,42]. Asa also plays a critical role in maintaining essential plant processes, such as photosynthesis, cell wall expansion, plant hormone production, regulation of antioxidant systems, and ion uptake, thereby increasing yield [43]. In vivo and in vitro studies have provided evidence that Asa can promote Fe solubility in mammals [44,45].

Several studies have investigated the use of Asa to promote plant growth. For instance, Shokr and Abdelhamid [46] found that foliar spraying of Asa resulted in improved vegetative growth and yield, and increased soluble solid substances and pigment constituents in pea (*Pisum sativum* L) plants compared to untreated controls (tap water). Similar results were observed in lettuce (*Lactuca sativa* L.) by Shafeek et al. [47] and in soybean (*Glycine max* L var. klark) by Mansour [48], where foliar spray of Asa led to significant increases in growth, yield, and nutritional content in the leaves. Ramírez et al. [49] reported that supplementing Arabidopsis seedlings with Asa had protective effects against Fe deficiency, maintaining chlorophyll content without increasing the internal Fe concentration. The authors suggested that this could be due to the antioxidant properties of Asa, as the protective effect was correlated with decreased levels of reactive oxygen species (ROS) and higher activity of ascorbate peroxidase, thereby maintaining cell redox homeostasis in Fe-deficient plants. In the present study, pre-treatment with Asa resulted in improved Fe utilization from the NHs by cucumber plants, similar to Cit pre-treatment, with lower root FCR activity (especially in case of NH-S2) compared to dFe plants.

Glycine is a commonly used amino acid in plant nutrition and is often utilized in the production of various amino-chelate fertilizers. External application of Gly can enhance the nitrogen status and concentration of mineral elements in plants [50]. Souri et al. [51] demonstrated that both foliar and soil application of Fe-glycine chelate resulted in a substantial improvement in the growth, yield, and quality characteristics of bean plants (*Phaseolus vulgaris* L.), even more so than Fe-EDDHA. Similarly, the application of Gly through Hoagland nutrient solution significantly increased leaf SPAD value and the fresh and dry weights of shoots and roots in coriander (*Coriandrum sativum*) plants, as reported in [52]. Noroozlo et al. [53] also showed that foliar application of Gly led to a significant increase in leaf Fe concentration compared to control Romain lettuce (*Lactuca sativa* subvar Sahara) plants. The latest study by Xu et al. [54] provided evidence using dual-isotope labeling tests that the presence of nitrogen in glycine can negatively influence Zn absorption by the leaves of waxy corn (*Zea mays* L. var. *ceratina* Kulesh), despite the fact that ZnGly facilitated the storage of Zn in the seeds, with improved Zn use efficiency. In the current study, pre-treatment with Gly improved the Fe utilization efficiency from the NHs in cucumber plants, as indicated by the increase in chlorophyll content and photosynthetic efficiency. However, the pH values of the nutrient solutions remained in the range of dFe plants for all NHs and the root FCR activity decreased only in case of NH-S2 and NH-S3, whereas it remained high for NH-S1.

The utilization of Fe in the mesophyllum tissues of treated leaves was also examined using XRF spectroscopy. The highest intensity was found in the case of macroelements such as Ca and K. Ca distribution in the leaf is determined by its accumulation in the cell walls and apoplastic spaces, especially around stomatal guard cells [55]. This was clearly seen in sFe plants and those treated with the three NHs in nutrient solutions. However, the Ca distribution in plants treated with NHs through the leaves resembled that of dFe plants. K distribution was mostly confined to the main veins, as it is delivered to leaves through the xylem in high concentrations. This provides a perfect orientation in leaf surface images, which is particularly useful in analyses of trace elements in low concentrations, such as Fe [56].

In the case of NHs applied in nutrient solution, a healthy delivery of Fe was found as compared to its distribution in sFe plants. Certainly, the intensity was much lower as the time of Fe supply was much shorter. In case of foliar application, NH-S1/Cit, NH-S1/Gly, NH-S2/Cit, and NH-S2/Asa also resulted in Fe distribution similar to that in sFe plants, as the veins were identified and the interveinal sections also had homogeneous Fe intensity. In the case of NH-S3, Fe intensity was much less confined to the veins in Cit and Asa pre-treatments. As the Fe in foliar treatment of dFe plants was not delivered from the nutrient solution, we suggest that Fe intensity confined to the veins may indicate its accumulation in sieve elements for distribution in the plant.

## 4. Materials and Methods

### 4.1. Plant Material

*Cucumis sativus* L. cv. Joker seeds were used. Seeds were first germinated on wet filter paper in darkness at 30 °C for 2 days, followed by 24-hour incubation with 0.5 mM CaSO_4_ solution in darkness to stimulate elongation of the hypocotyl. The seedlings were then transferred to a modified quarter-strength unbuffered Hoagland solution containing 1.25 mM KNO_3_, 1.25 mM Ca(NO_3_)_2_, 0.5 mM MgSO_4_, 0.25 mM KH_2_PO_4_, 11.6 μM H_3_BO_3_, 4.5 μΜ MnCl_2_, 0.19 μM ZnSO_4_, 0.12 μM Na_2_MoO_4_, and 0.08 μM CuSO_4_ (Fe-deficient plants). Fe-sufficient plants were grown in the same solution but with 10 μM Fe (III)-citrate. Another set of plants was grown in nutrient solution buffered to pH 8.5. In order to maintain the desired pH value, 1 mM KHCO_3_ and 0.3 g CaCO_3_ were added to each pot. In this case, Fe-sufficient plants were grown with Fe (III)-EDDHA. Each plant was grown in a single plastic pot with 400 mL nutrient solution, which was replaced every other day for 2 weeks. The plants were grown in a climate-controlled growth chamber with a 120 μmol s^−1^m^−2^ photosynthetic photon flux density and 14/10 h (light/dark) photoperiod, with temperatures of 20–26 °C and a relative humidity of approximately 70/75%. The experiments were conducted using 2-week-old Fe-deficient plants.

### 4.2. Treatment with Nanomaterials

Three NH colloid suspensions were applied in the experiments, which were prepared together in one synthesis process investigated and described in our previous study [26]. The different suspensions characterized by transmission electron microscopy and X-ray diffraction were originally ferrihydrite with a particle size of 4–7 nm (PEG-1500 coating, NH-S1), nanohematite with a particle size of 12–25 nm (Emulsion 104D coating, NH-S2), and nanoferrihydrite/nanohematite with a particle size of 4–8 nm (SOLUTOL HS 15 coating, NH-S3) [26]. With aging there was a slow conversion of the particles towards nanohematite components. The NHs were stabilized with different coating agents: PEG-1500 (NH-S1), Emulsion 104D (NH-S2), and SOLUTOL HS 15 (NH-S3). The nanomaterial treatments were applied to Fe-deficient plants in nutrient solution or as foliar fertilizer. In the nutrient solutions, the equivalent amount of NH suspension was added to give a 20 µM nominal concentration of Fe. In the foliar treatment, the NH suspensions were applied to the 2nd leaves of plants twice a day for 2 days in a nominal concentration of 2 mM Fe. Prior to foliar spray treatment with the NH suspensions, the leaves were sprayed with Cit, Asa, or Gly solution at a final concentration of 0.5 mM, prepared in 0.02% nonit as a surfactant.

### 4.3. Physiological Parameters

Chlorophyll content was estimated non-destructively using a SPAD-502+ Chlorophyll Meter device (Konica-Minolta, Osaka, Japan) for 4 days starting from the day of treatment in both cases.

Three days post-treatment, disks were cut from the 2nd leaves using a cork borer and the chlorophyll was extracted in 80% acetone (V/V) and 5 mM Tricin buffer (pH 7.5) using a mortar and pestle. After centrifugation at 10,000× *g* for 5 min, the chlorophyll concentration was measured using a UV2101 spectrophotometer (Shimadzu, Kyoto, Japan) with the extinction coefficients of Porra et al. [57].

Total dry weight of the plants was determined after drying at 80 °C for 2 days.

Gas exchange and photosynthetic activity were measured using an LI-6800F portable photosynthesis system (LICOR Biosciences, Lincoln, NE, USA) with a 2 cm^2^ aperture for leaf samples. The applied parameter settings were as follows: CO_2_ concentrations of 400 µmol mol^−1^ air, relative humidity of 60%, with a flow rate of approximately 600 µmol s^−1^. To induce Chl *a* fluorescence, leaves were dark-adapted in the leaf chamber of the device until reaching a stable fluorescence signal in approx. 20 min. An actinic radiation of 600 µmol photons m^−2^ s^−1^ PPFD was applied. Measurements of different photosynthetic parameters (A_net_, Fv/Fm, Fv’/Fm’, ΦPSII, ΦCO_2_, and NPQ) were performed between 9 a.m. and 5 p.m. on the 2nd leaves 3–4 days post-treatment. The parameters used in this study were defined and calculated using the instrument’s internal software: https://www.licor.com/documents/ajncmgt9xtonajwvs3n6hxxw5u9dlfai (accessed on 5 August 2023).

### 4.4. X-ray Fluorescence Imaging

Once the treatment was completed, the 2nd leaves were collected and dried at 60 °C under press, ensuring a smooth surface of the leaf blade. X-ray fluorescence (XRF) analysis was conducted using an XGT-7200 V analytical imaging instrument (Horiba, Osaka, Japan) equipped with a Rh X-ray tube and silicon drift detector (SDD). The sample was positioned in the measuring chamber at atmospheric pressure and room temperature. Acceleration voltage and current of 50 kV and 1 mA, respectively, and an X-ray guide tube of 100 μm were employed. The element distribution maps were prepared using the characteristic Kα photons emitted by the Fe (6.405 keV), Ca (3.692 keV), and K (8.637 keV) atoms and detected by the SDD. A mapping area of 15.36 × 15.36 mm was analyzed, with 1000 s survey time per frame, and data were collected 5 times per pixel.

### 4.5. Ferric Chelate Reductase Assay

The ferric chelate reductase assay was conducted according to the method used by Kovács et al. [58]. Spectrophotometric measurement of the absorbance of the [Fe(II)-bathophenanthroline disulfonate_3_]^4−^ complexes was performed at a wavelength of 535 nm using a Shimadzu UV-2101PC spectrophotometer (Shimadzu, Kyoto, Japan). The extinction coefficient of the complex used for calculations was 22.14 mM^−1^ cm^−1^, as reported by Smith et al. [59].

### 4.6. Element Analysis

Element concentration estimation was performed after acid digestion of dried plant samples. The samples were dried for two days at 85 °C and then digested in ccH_2_O_2_ for 1 h, followed by the addition of ccHNO_3_, and incubation for 15 min at 60 °C and 45 min at 120 °C. The resulting solution was filtered through MN 640 W filter paper (Macherey-Nagel, Düren, Germany) and thoroughly homogenized. The filtrate was then analyzed for elemental content using an inductively coupled plasma–optical emission spectrometer (ICP-OES) (Spectro Genesis, SPECTRO, Freital, Germany) with the help of a multi-element standard for 33 elements (Loba Chemie Product code: I166N, Loba Chemie PVT, Mumbai, India) used for calibration.

### 4.7. Statistical Treatment

The experiments were performed twice. To analyze differences, ANOVA was performed with the Tukey-Kramer multiple comparison post hoc test using InStat v. 3.00 software (GraphPad Software, Inc., San Diego, CA, USA). A significant difference was considered when the similarity of the samples was less than *p* = 0.05.

## 5. Conclusions

In our study, three NH colloid suspensions with different coatings were applied as nanofertilizers through nutrient solution or as foliar spray to young dFe cucumber plants. The results demonstrated that the NHs effectively improved various physiological parameters and overall growth when administered via both routes, underscoring their efficacy as nanofertilizers. However, it was noted that the NHs did not yield significant growth and physiological improvement when applied in nutrient solution at alkaline pH levels, as none of the coating materials were effective in maintaining continuous Fe supply. The introduction of three different pre-treatments (Cit, Asa, and Gly) prior to foliar spraying remarkably facilitated Fe uptake by the leaves, leading to enhanced photosynthetic parameters and reduced root ferric chelate reductase activity, indicating the downregulation of Fe efficiency reactions. The different coatings did not result in significantly different micronutrient efficiency; they performed equally well. Overall, the NHs assessed in this study hold promise as valuable sources of Fe, offering a potential solution to address Fe deficiency in agricultural settings.

## Figures and Tables

**Figure 1 plants-12-03104-f001:**
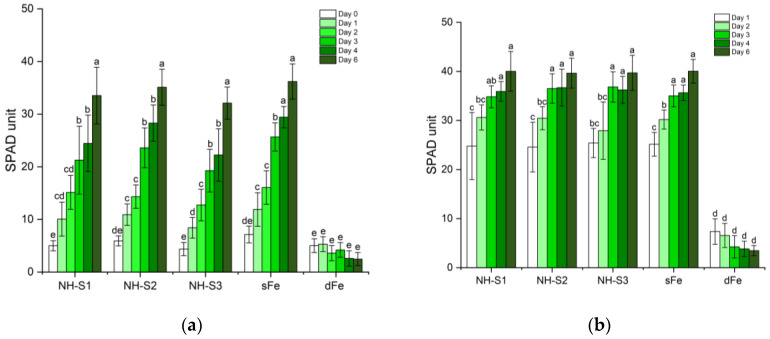
SPAD values of the 1st and 2nd leaves of dFe plants supplied with NH-S1-3 in unbuffered nutrient solution (**a**,**b**) and as foliar treatment combined with citric acid (Cit), ascorbic acid (Asa), or glycine (Gly) pre-treatment (**c**,**d**) (dFe and sFe plants served as controls). Data are shown as mean ± SD (*n* = 5). Significant differences between data are shown by different letters, *p* < 0.05.

**Figure 2 plants-12-03104-f002:**
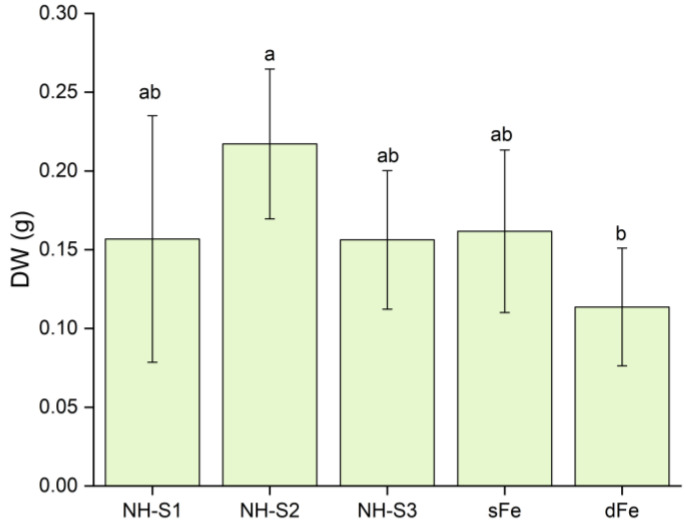
Total dry weight (DW) of plants grown in unbuffered nutrient solutions with 3 different NHs after 6 days. Data are presented as mean ± SD (*n* = 5) significant differences between data are shown by different letters, *p* < 0.05.

**Figure 3 plants-12-03104-f003:**
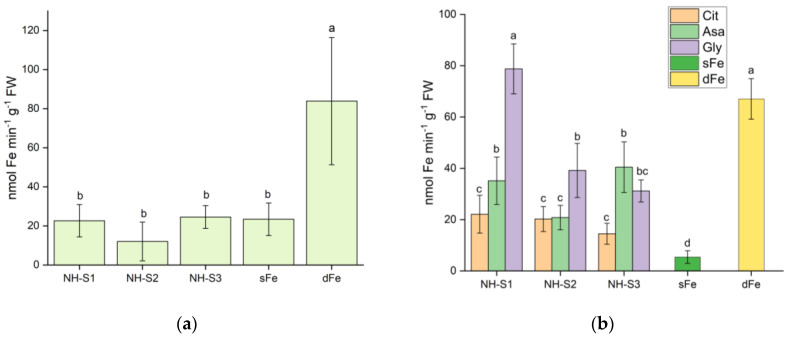
Ferric chelate reductase activity of the root tips 3 days after supplying NH-S1-3 to dFe plants in (unbuffered) nutrient solution (**a**) or as foliar treatment (**b**). Data are presented as mean ± SD (*n* = 5). Significant differences between data are shown by different letters, *p* < 0.05.

**Figure 4 plants-12-03104-f004:**
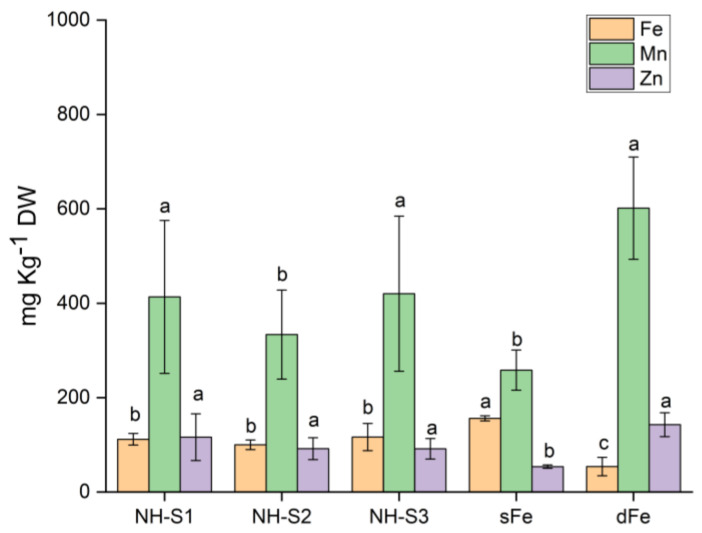
Element analysis of the 2nd leaves of plants treated with NH-S1-3 supplied in nutrient solution after 6 days. dFe and sFe plants served as controls. Data are presented as mean ± SD (*n* = 5). Statistical analysis was run independently for different elements. Significant differences between data are shown by different letters, *p* < 0.05.

**Figure 5 plants-12-03104-f005:**
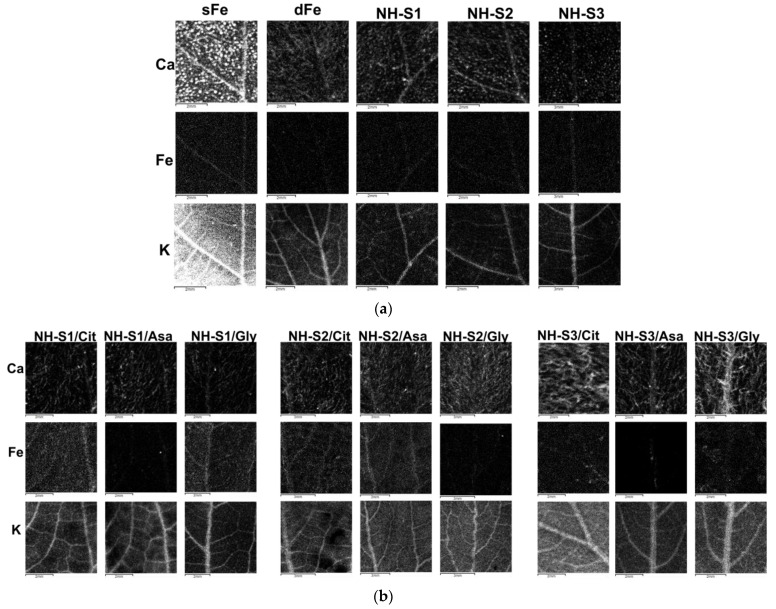
Representative X-ray fluorescence maps of arbitrarily selected leaf sections from the 2nd leaves of plants harvested 3 days following foliar treatment. Panel (**a**) demonstrates leaves of control plants (sFe and dFe) and plants treated with NH-S1-3 supplied in nutrient solution, while panel (**b**) exhibits leaves of plants treated with NH-S1-3 applied as foliar spray and pre-treated with Cit, Asa, or Gly. Data for optical images were collected for about 150 min at Kα emission for Ca, Fe, and K (peak emissions are 3.69, 6.40, and 3.31 keV, respectively).

**Table 1 plants-12-03104-t001:** The chlorophyll content of dFe plants treated with NH-S1-3 supplied in nutrient solutions at two different pH values. Data are shown as mean ± SD (*n* = 5). Significant differences between data are shown by different letters, *p* < 0.05.

Treatment	Chl a+b (µg/g FW)		Chl a/b	
	Acidic pH	Alkaline pH	Acidic pH	Alkaline pH
NH-S1	1924.8 ± 287.3 ^a^	171.9 ± 28.99 ^b^	3.0 ± 0.05 ^b^	3.0 ± 0.22
NH-S2	2063.9 ± 76.3 ^a^	173.8 ± 28.30 ^b^	3.0 ± 0.09 ^b^	3.0 ± 0.57
NH-S3	2092.9 ± 171.8 ^a^	168.2 ± 39.80 ^b^	2.9 ± 0.03 ^b^	2.7 ± 0.28
sFe	2555.28 ± 191.76 ^a^	2273.06 ± 246.92 ^a^	2.4 ± 0.14 ^b^	2.8 ± 0.03
dFe	92.2 ± 27.7 ^b^	162.6 ± 70.97 ^b^	4.9 ± 1.03 ^a^	2.8 ± 0.54

**Table 2 plants-12-03104-t002:** The pH values of nutrient solutions 3 days after supplying NH-S1-3 to dFe plants in (unbuffered) nutrient solution or as foliar treatment. The initial pH values of the fresh nutrient solutions were 4.91 for dFe, 5.50 for sFe, 4.81 for NH-S1, 4.71 for NH-S2, and 4.66 for NH-S3. Data are presented as mean ± SD (*n* = 5). Significant differences between data are shown by different letters, *p* < 0.05, where statistical analysis was run independently for hydroponic and foliar treatments.

Treatment	Root Supply	Foliar Supply		
		Cit	Asa	Gly
NH-S1	4.27 ± 0.20 ^B^	4.57 ± 0.03 ^ab^	4.36 ± 0.19 ^b^	4.04 ± 0.16 ^b^
NH-S2	4.39 ± 0.25 ^AB^	4.46 ± 0.17 ^b^	4.13 ± 0.13 ^b^	3.84 ± 0.15 ^b^
NH-S3	4.37 ± 0.26 ^AB^	4.60 ± 0.22 ^ab^	4.89 ± 0.05 ^ab^	4.18 ± 0.35 ^b^
sFe	6.29 ± 0.08 ^A^	5.97 ± 0.20 ^a^		
dFe	3.88 ± 0.17 ^C^	4.07 ± 0.36 ^b^		

**Table 3 plants-12-03104-t003:** Photosynthetic and Chl a fluorescence induction parameters of plants after 3 days of supplementation with NH-S1-3 (CO_2_ assimilation, Anet; maximum quantum efficiency of PSII, Fv/Fm; actual quantum efficiency of PSII in light-adapted state, Fv’/Fm’; actual quantum efficiency of PSII, ΦPSII; quantum yield of CO2, ΦCO_2_; non-photochemical quenching, NPQ). The NHs were applied either in nutrient solution or as foliar treatment after pre-treatment with Cit, Asa, or Gly. Untreated dFe and sFe plants served as controls. Data are presented as mean ± SD (*n* = 5). Statistical analysis was run independently for hydroponic and foliar treatments. Significant differences between data are shown by different letters, *p* < 0.05.

Root Supply	Assimilation (A_net_) (µmol CO_2_ m^−2^ s^−1^)	Fv/Fm	Fv’/Fm’	ΦPSII	ΦCO_2_ (μmol CO_2_ μmol^−1^ Photons)	NPQ
NH-S1	12.901 ± 2.526 ^a^	0.798 ± 0.005 ^a^	0.634 ± 0.034 ^a^	0.478 ± 0.031 ^a^	0.033 ± 0.006 ^A^	0.914 ± 0.242 ^a^
NH-S2	14.840 ± 1.426 ^a^	0.798 ± 0.001 ^a^	0.641 ± 0.025 ^a^	0.488 ± 0.029 ^a^	0.036 ± 0.003 ^A^	0.832 ± 0.173 ^a^
NH-S3	14.251 ± 0.560 ^a^	0.799 ± 0.004 ^a^	0.637 ± 0.001 ^a^	0.487 ± 0.010 ^a^	0.033 ± 0.002 ^A^	0.921 ± 0.124 ^a^
sFe	14.156 ± 3.421 ^a^	0.806 ± 0.004 ^a^	0.650 ± 0.027 ^a^	0.491 ± 0.046 ^a^	0.033 ± 0.007 ^A^	0.845 ± 0.130 ^a^
dFe	−0.506 ± 0.123 ^b^	0.570 ± 0.047 ^b^	0.389 ± 0.125 ^b^	0.090 ± 0.009 ^b^	0.003 ± 0.003 ^b^	0.574 ± 0.130 ^b^
Foliar supply						
NH-S1						
Cit	9.596 ± 2.374 ^ab^	0.805 ± 0.002 ^a^	0.568 ± 0.007 ^a^	0.388 ± 0.015 ^ab^	0.025 ± 0.004 ^a^	1.351 ± 0.105 ^a^
Asa	10.725 ± 3.810 ^ab^	0.807 ± 0.006 ^a^	0.587 ± 0.045 ^a^	0.426 ± 0.041 ^ab^	0.028 ± 0.006 ^a^	1.309 ± 0.317 ^a^
Gly	11.193 ±3.895 ^ab^	0.809 ± 0.002 ^a^	0.593 ± 0.015 ^a^	0.426 ± 0.031 ^ab^	0.029 ± 0.008 ^a^	1.204 ± 0.128 ^a^
NH-S2						
Cit	6.423 ± 0.356 ^b^	0.772 ± 0.002 ^a^	0.532 ± 0.042 ^a^	0.324 ± 0.046 ^b^	0.024 ± 0.004 ^a^	1.111 ± 0.199 ^a^
Asa	9.618 ± 2.917 ^ab^	0.777 ± 0.008 ^a^	0.586 ± 0.047 ^a^	0.401 ± 0.089 ^ab^	0.026 ± 0.004 ^a^	1.196 ± 0.447 ^a^
Gly	10.241 ± 1.368 ^ab^	0.776 ± 0.008 ^a^	0.611 ± 0.026 ^a^	0.434 ± 0.017 ^a^	0.035 ± 0.011 ^a^	0.867 ± 0.079 ^a^
NH-S3						
Cit	4.835 ± 1.478 ^b^	0.790 ± 0.009 ^a^	0.579 ± 0.038 ^a^	0.286 ± 0.047 ^b^	0.020 ± 0.005 ^a^	1.082 ± 0.387 ^a^
Asa	6.308 ± 1.163 ^b^	0.777 ± 0.009 ^a^	0.596 ± 0.062 ^a^	0.319 ± 0.033 ^b^	0.023 ± 0.003 ^a^	0.941 ± 0.501 ^a^
Gly	5.999 ± 1.689 ^b^	0.776 ± 0.013 ^a^	0.588 ± 0.036 ^a^	0.322 ± 0.046 ^b^	0.022 ± 0.005 ^a^	0.972 ± 0.320 ^a^
sFe	14.770 ± 3.799 ^a^	0.797 ± 0.006 ^a^	0.637 ± 0.031 ^a^	0.463 ± 0.051 ^a^	0.040 ± 0.006 ^a^	0.821 ± 0.186 ^a^
dFe	−1.950 ± 0.656 ^c^	0.473 ± 0.051 ^b^	0.313 ± 0.076 ^b^	0.043 ± 0.006 ^c^	0.004 ± 0.001 ^b^	0.462 ± 0.187 ^b^

## Data Availability

The data presented in this study are available on reasonable request from the corresponding author.

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
