# Peer review of "Coated Hematite Nanoparticles Alleviate Iron Deficiency in Cucumber in Acidic Nutrient Solution and as Foliar Spray"

_plants, 2023, doi:10.3390/plants12173104_

Round 1

Reviewer 1 Report

In this work, hematite nanoparticles with different coatings were used to tackle the problem of iron deficiency in cucumber plants. The manuscript is well-written and I don't find major flaws. I suggest the following things for improvement:

Abstract, toward the end: ‘via both modes of application’, ‘through both modes of application’ occurs in 2 consecutive phrases. Try to avoid repetitions. Use them only when it can’t be done otherwise.

Line number 73: ‘thus rendering their effectiveness’. Not clear. Rephrase. You mean ‘decreasing’?

Line 114: So which are those 3 different surfactants? Specify their names, please! You write it later in the text, but it is better to write it earlier, I think.

Could you show TEM images and XRD measurements of your  hematite nanoparticles? It is helpful, even though you used a protocol from your previous work. If you have extra TEM images that are not identical with those published at Ref 26, you can show them.

References list:

Ref 12 is the same with Ref 37. So please remove the Ref 37 and re-arrange the numbering of the references in both text and list of references.

Ref 40: article or page number is missing.

Ref 41: You need to add the article number, not the page range.

Ref 55: Page or article number is missing.

Author Response

Answers to Reviewer 1:

Abstract, toward the end: ‘via both modes of application’, ‘through both modes of application’ occurs in 2 consecutive phrases. Try to avoid repetitions. Use them only when it can’t be done otherwise.

Answer: Thank you for pointing out the repetition. The repetitive sentences have been deleted.

Line number 73: ‘thus rendering their effectiveness’. Not clear. Rephrase. You mean ‘decreasing’?

Answer: Sorry for the misleading phrase: ’rendering’ has been replaced with ’decreasing’

Line 114: So which are those 3 different surfactants? Specify their names, please! You write it later in the text, but it is better to write it earlier, I think.

Answer: Thank you for this suggestion. The 3 different surfactants are now specified at the beginning of Results as follows: „In this study three NH colloid suspensions differing mainly in surfactant (PEG-1500, NH-S1; Emulsion 104D, NH-S2; SOLUTOL HS 15, NH-S3) for stabilization were used as amendments to the nutrient solution or as foliar fertilizer.”

Could you show TEM images and XRD measurements of your  hematite nanoparticles? It is helpful, even though you used a protocol from your previous work. If you have extra TEM images that are not identical with those published at Ref 26, you can show them.

Answer: In this study, we have actually used the nanomaterials that we have synthesized for our previous study. In Gracheva et al. 2022 (Ref 26) the three NHs applied were only characterised but not tested in plant experiments. The TEM and XRD characterisation was already done and the best images were published in that paper. (Furthermore contributing authors preparing that part of the previous study are not involved in the present one just to show alternate images of the same measurement.) Although we did not provide further images in the present manuscript but we have now supplied more detailed information in the Materials and methods based on Ref 26.

References list:

Ref 12 is the same with Ref 37. So please remove the Ref 37 and re-arrange the numbering of the references in both text and list of references.

Ref 40: article or page number is missing.

Ref 41: You need to add the article number, not the page range.

Ref 55: Page or article number is missing.

Answer: The reference list has been checked and corrected

We are grateful to the reviewer for pointing out the above weaknesses of our manucript and we hope that our corrections and answers are acceptable.

Reviewer 2 Report

The study examined the efficiency of different hematite nanocolloid suspensions with surfactant coatings as a supplement to nutrient solutions and as foliar sprays to address Fe deficiency and identify the best application technique to promote plant growth and reduce the effects of Fe shortage on cucumber plants. The manuscript is well written and structured, and it has been conducted with an appropriate methodology. Moreover, the authors have explained the results of the experiment substantially. I believe that the findings in this study warrant publication in the Plants Journal. Therefore, I would like to suggest the acceptance of the manuscript for publication with minor revisions, according to the comments and suggestions below:

·      1.       Suggestion to consider revising Figure 1. It shows post hoc tests for days 6 (Figs. 1a, 1b) and 3 (Figs. 1c, 1d), and it is not clear whether there was a significant difference in other days and among treatments. My suggestion is to use a line graph with days on the X axis and show different treatments with legends instead of Days. In this case, we can see the response over time and test whether it is linear, quadratic, or exponential.

2.       Check for typographical errors (e.g., lines 468–470, starting the sentence with number 3 and lacking and in between mortar and pestle).

3.       Check for consistency in writing the references (e.g., #21, 48, and 49).

The study examined the efficiency of different hematite nanocolloid suspensions with surfactant coatings as a supplement to nutrient solutions and as foliar sprays to address Fe deficiency and identify the best application technique to promote plant growth and reduce the effects of Fe shortage on cucumber plants. The manuscript is well written and structured, and it has been conducted with an appropriate methodology. Moreover, the authors have explained the results of the experiment substantially. I believe that the findings in this study warrant publication in the Plants Journal. Therefore, I would like to suggest the acceptance of the manuscript for publication with minor revisions, according to the comments and suggestions below:

·   1.       Suggestion to consider revising Figure 1. It shows post hoc tests for days 6 (Figs. 1a, 1b) and 3 (Figs. 1c, 1d), and it is not clear whether there was a significant difference in other days and among treatments. My suggestion is to use a line graph with days on the X axis and show different treatments with legends instead of Days. In this case, we can see the response over time and test whether it is linear, quadratic, or exponential.

2.       Check for typographical errors (e.g., lines 468–470, starting the sentence with number 3 and lacking and in between mortar and pestle).

3.       Check for consistency in writing the references (e.g., #21, 48, and 49).

Author Response

Answers to Reviewer 2:

  1. Suggestion to consider revising Figure 1. It shows post hoc tests for days 6 (Figs. 1a, 1b) and 3 (Figs. 1c, 1d), and it is not clear whether there was a significant difference in other days and among treatments. My suggestion is to use a line graph with days on the X axis and show different treatments with legends instead of Days. In this case, we can see the response over time and test whether it is linear, quadratic, or exponential.

Answer: Thank you for the suggestion. Previously we have tried to prepare a line graph but there were so many overlapping points that we found it inappropriate and decided to apply a bar graph instead. We have performed the full statistical analysis and now the letters are provided in the revised figure.

Concerning the response of plants in time, it can also be seen in the bar graph that in most cases there is no significant difference between the treatments except for NH-S1 (2nd leaf, foliar treatment) for which the greening somewhat accelerated between day1 and day2. This may be due to the smallest particle size of this NH preparation. But as SPAD index highly depends on leaf thickness and the result may also be due to delayed leaf expansion. For this reason we would not draw important conclusions based on SPAD values and only intended to demonstrate the continuous greening process in both leaves.

  1. Check for typographical errors (e.g., lines 468–470, starting the sentence with number 3 and lacking and in between mortar and pestle).

Answer: Thank you for pointing out the typo. It has been corrected.

  1. Check for consistency in writing the references (e.g., #21, 48, and 49).

Answer: The reference list has been checked and corrected.

 We are grateful to the reviewer for pointing out the above weaknesses of our manucript and we hope that our corrections and answers are acceptable.